# Association between maternal vegetable intake during pregnancy and allergy in offspring: Japan Environment and Children's Study

Kohei Ogawa[1,2,3]*, Kyongsun Pak[4], Kiwako Yamamoto-Hanada[3,5], Kazue Ishitsuka[3], Hatoko Sasaki[3], Hidetoshi Mezawa[3], Mayako Saito-Abe[3,5], Miori Sato[3,5], Limin Yang[3], Minaho Nishizato[3], Mizuho Konishi[3], Haruhiko Sago[1], Yukihiro Ohya[3,5]*, on behalf of Japan Environment and Children's Study (JECS) Group[¶]

1 Center for Maternal-Fetal, Neonatal and Reproductive Medicine, National Center for Child Health and Development, Tokyo, Japan, 2 Department of Social Medicine, National Research Institute for Child Health and Development, Tokyo, Japan, 3 Medical Support Center for the Japan Environment and Children's Study, National Center for Child Health and Development, Tokyo, Japan, 4 Division of Biostatistics, Department of Data Management, Center for Clinical Research, National Center for Child Health and Development, Tokyo, Japan, 5 Allergy Center, National Center for Child Health and Development, Tokyo, Japan

¶ Membership of the Japan Environment and Children's Study (JECS) Group is listed in the Acknowledgments.
* ogawa-k@ncchd.go.jp (KO); ohya-y@ncchd.go.jp (YO)

**Data Availability Statement:** Data are unsuitable for public deposition due to ethical restrictions and legal framework of Japan. It is prohibited by the Act

## Abstract

The association between maternal diet during pregnancy and allergy in offspring remains contentious. Here, we examined the association between maternal intake of vegetables and related nutrients during pregnancy and allergic diseases in offspring at one year of age. A cohort of 80,270 pregnant women enrolled in the Japan Environment and Children's Study were asked to respond to a food frequency questionnaire during pregnancy and the International Study of Asthma and Allergies in Childhood questionnaire at one year postpartum. The women were categorized into quintiles according to the energy-adjusted maternal intake of vegetables and related nutrients. Using the categorizations as exposure variables, the adjusted odds ratios (aORs) with 95% confidence intervals (CIs) were determined for the allergic outcomes, including asthma, wheeze, atopic dermatitis, eczema, and food allergy, in the offspring per quintile at one year of age. Of the 80,270 participants, 2,027 (2.5%), 15,617 (19.6%), 3,477 (4.3%), 14,929 (18.7%), 13,801 (17.2%), and 25,028 (31.3%) children experienced asthma, wheeze, atopic dermatitis, eczema, food allergy, and some form of allergic disease, respectively. The aORs of each quintile of maternal vegetable intake for all allergic outcomes were close to 1.0 compared to the lowest quintile. The lowest aOR was found in the association of maternal cruciferous vegetable intake with asthma (aOR: 0.82, 95% CI: 0.70–0.96) and highest was found in the association of maternal total vegetable intake with atopic dermatitis (aOR: 1.17, 95% CI: 1.04–1.31). The risk of allergic outcomes for the various nutrients related to vegetable consumption was close to 1.0. The maternal intake of vegetables and various related nutrients during pregnancy had little or no

on the Protection of Personal Information (Act No.57 of 30 May 2003, amendment on 9 September 2015) to publicly deposit the data containing personal information. Ethical Guidelines for Epidemiological Research enforced by the Japan Ministry of Education, Culture, Sports, Science and Technology and the Ministry of Health, Labour and Welfare also restricts the open sharing of the epidemiologic data. All inquiries about access to data should be sent to: jecs-en@nies.go.jp. The person responsible for handling enquiries sent to this e-mail address is Dr Shoji F. Nakayama, JECS Programme Office, National Institute for Environmental Studies.

**Funding:** The JECS was funded by the Ministry of the Environment, Japan. The findings and conclusions of this article are solely the responsibility of the authors and do not represent the official views of the Japanese government.

**Competing interests:** All the authors declare that they have no conflicts of interest associated with the publication of this research.

association with any of the allergic outcomes, including asthma, wheezing, atopic dermatitis, eczema, and food allergy, in offspring at one year.

## Introduction

To date, numerous risk factors of allergic disease in childhood, including atopic disease, eczema, wheeze, and asthma, have been identified, such as second-hand smoking, family history of atopic disease, and viral infections during infancy [1, 2]. Maternal diet during pregnancy has been considered to play an important role in the development of allergic disease in offspring [3], and several epidemiological studies on this topic have been conducted [4–6].

Maternal vegetable intake during pregnancy is an interesting exposure category due to the anti-oxidative and anti-inflammatory properties of vegetables, which may be related to the development of allergic diseases [7, 8]. Although several studies of maternal vegetable intake during pregnancy have investigated its association with the development of allergic disease in offspring using a self-reported database, they have yielded inconsistent results. For instance, while two studies showed a significant inverse association between maternal vegetable intake and wheeze in offspring [9, 10], three studies showed a non-significant association [11–13]. Similarly, while one study demonstrated a significant inverse association between yellow and green vegetable intake during pregnancy and the occurrence of eczema in childhood [11], two studies had null findings [13, 14]. One possible explanation for this inconsistency is the relatively small sample sizes (ranging from 310 to 3086) used in these observational studies. Thus, a study with a large sample size is necessary to suggest which of the previous findings is valid. Furthermore, the association between the maternal intake of nutrients related to vegetables and the incidence of childhood allergies has not been widely studied [4] despite its importance.

The current study used a large cohort database to determine whether an association exists between the maternal intake of vegetables and related nutrients during pregnancy and the development of allergic diseases in offspring.

## Materials and methods

### Study population

This prospective, longitudinal study was based on the Japan Environment and Children's Study (JECS), a nationwide, prospective birth cohort study conducted in Japan [15–18]. Pregnant women who participated in the JECS were recruited between January 2011 and March 2014 from 15 study regions that covered most of Japan. Women who agreed to participate were asked to complete a questionnaire to gather their demographic data, including socioeconomic status (SES), medical history, anthropometry, and dietary information. A self-administered food frequency questionnaire (FFQ), which was completed during pregnancy (as a general rule, from 22+0 to 27+6), was used to assess the maternal diet during pregnancy. After delivery, the participants responded every six months to another questionnaire that asked about allergic symptoms in their children. Birth outcomes and complications related to pregnancy were separately collected from medical records. All the analyses were based upon the "jecs-an-20180131" data set, which was created in June 2016 and revised in October 2016. The data obtained during pregnancy and one year postpartum in the JECS were then analyzed. Women with a singleton delivery without congenital malformations were included. Women who were missing values on every outcome variable or failed to supply information about their

vegetable intake on the FFQ were excluded (those missing one or two values were kept in our study population). Of those, our primary analysis was conducted based on the dataset with complete data on each variable, as our dataset contained a substantial sample. The results were subsequently confirmed by a sensitivity analysis using multiple imputations based on the subset with missing data. The JECS protocol was approved by the review board for epidemiological studies of the Ministry of the Environment, Japan, and by the ethics committees of all participating institutions. These institutions include the National Institute for Environmental Studies (which leads the JECS), the National Center for Child Health and Development, Hokkaido University, Sapporo Medical University, Asahikawa Medical College, Japanese Red Cross Hokkaido College of Nursing, Tohoku University, Fukushima Medical University, Chiba University, Yokohama City University, University of Yamanashi, Shinshu University; University of Toyama, Nagoya City University, Kyoto University, Doshisha University, Osaka University, Osaka Medical Center and Research Institute for Maternal and Child Health, Hyogo College of Medicine, Tottori University, Kochi University, University of Occupational and Environmental Health, Kyushu University, Kumamoto University, University of Miyazaki, and the University of the Ryukyus. The JECS was conducted in accordance with the guidelines of the Declaration of Helsinki and other nationally valid regulations [19]. Written informed consent was obtained from each participant.

## Measures

**FFQ.** We assessed maternal dietary intake during pregnancy using an FFQ. The FFQ was completed during mid- to late-pregnancy, and other profiles of the FFQ have been described elsewhere in detail [20]. Validation of the FFQ was conducted in previous studies using dietary records for three days and blood samples as a reference for the Japanese population [21–23], although validation specifically for pregnant women has not been done. The intake of energy, nutrients, and food groups was estimated using a food composition table developed for the FFQ based on the 2010 edition of the Standardized Tables of Food Composition in Japan [24].

Vegetables were categorized into subclasses according to the following definitions, according to a previous study [9]. Folate-rich vegetables were defined as vegetables containing 100 μg or more of folate per 100 g, such as spinach, green spring onion, Chinese chives, *Glebionis coronaria*, Japanese mustard spinach, broccoli, and asparagus. Green and yellow vegetables were defined as containing 600 μg or more of carotene per 100 g and included carrot, spinach, pumpkin, green pepper, tomato, green spring onion, Chinese chives, *Glebionis coronaria*, Japanese mustard spinach, broccoli, kidney bean, and asparagus. Cruciferous vegetables included cabbage, radish, Chinese cabbage, Japanese mustard spinach, and broccoli.

## Allergic outcomes

Information on allergic outcomes was obtained using a questionnaire when the offspring were one year old. Wheezing and eczema symptoms in the offspring at this age were assessed using a questionnaire, which was modified from the International Study of Asthma and Allergies in Childhood (ISSAC) as a validated questionnaire [25, 26]. The presence of wheeze was determined by a positive response on the ISSAC to the following question: "Has your child had wheezing or whistling in the chest in the past 12 months?" Similarly, the presence of eczema was determined by a positive response to the question: "Has your child had rashes with itching which improved, then worsened, in the past 12 months?" The prevalence of allergic diseases (asthma, atopic dermatitis, and food allergy) was assessed based on a self-reported doctor's diagnosis obtained via a questionnaire when the offspring were one year old. "Any allergy"

was defined as the presence of any allergic outcome, including asthma, wheeze, atopic dermatitis, eczema, and food allergy.

## Covariates

Covariates, including maternal age, pre-pregnancy height and weight, weight gain during pregnancy, parity, conception method (with or without assisted reproductive technology), pre-existing hypertension or diabetes, maternal allergic history, delivery mode, and infant sex, were retrieved from medical records. Other variables, such as the place of recruitment, parental smoking status, maternal SES, including maternal education (junior high school, senior high school, university), household income (<4 million-yen, ≥ 4 and <6 million-yen, ≥6 million-yen), maternal folic acid supplementation during pregnancy (yes or no), and breastfeeding at one month after delivery (breastfeeding only, mixed feeding, artificial mild feeding), were obtained via a questionnaire at baseline. The covariates were treated as confounding factors according to previous studies [27].

## Statistical analysis

Of the 90,422 women with a singleton delivery without a congenital malformation, 8,975 and 1,177 were excluded due to missing values for every outcome and no responses concerning vegetable intake on the FFQ, respectively. Thus, the remaining 80,270 women were included in our study population. Our main analysis was conducted after excluding those with missing data on confounding factors (n = 9,917) and each outcome (n for wheeze: 438, n for eczema: 387, n for any allergy: 311 [some of those overlapped with missing confounding factors]). Thus, the main analyses for asthma, wheeze, atopic dermatitis, eczema, food allergy, and any allergy were conducted for 70,353, 70,010, 70,353, 70,044, 70,353, and 70,103 people, respectively. A sensitivity analysis on 80,270 women using multiple imputations was conducted for the entire cohort (**Fig 1**).

First, the participants were categorized into quintiles according to the energy-adjusted maternal intake of each vegetable or its related nutrient after log-transformation. Q1 was the lowest quintile, and Q5 was the highest quintile. To adjust the intake amounts by energy, we used the residual method [28]. An example of a quintile based on the energy-adjusted estimate of the intake for total vegetables is shown in **Fig 1**. Second, we used crude and multivariable logistic regression analyses to estimate the crude and adjusted odds ratios (ORs and aORs, respectively) of the allergic outcomes between the quintiles of each exposure using Q1 as the reference category, with 95% confidence intervals (CI). For multivariate analysis, maternal age, place of recruitment, maternal height, pre-pregnancy body mass index (BMI), maternal weight gain during pregnancy, parity, conception method (assisted reproductive technology or not), pre-existing maternal hypertension, pre-existing maternal diabetes, parental allergic history, parental smoking, maternal education, maternal household income, infant gender, maternal folic acid supplementation during pregnancy, breastfeeding at one month after delivery, estimated maternal total energy intake during pregnancy, and delivery mode were adjusted. Third, for sensitivity analysis, the aORs were also assessed via multiple imputations by the chained equation (MICE) to confirm the robustness of our results. MICE was performed with 20 sets of complete data, including the confounding factors. For another sensitivity analysis, we assessed the association between maternal allergic history and maternal vegetable intake during pregnancy. Maternal allergic history was considered to be one of the most important potential confounding factors because some women with allergic history may be likely to begin consuming vegetables to avoid the development of allergic diseases in their offspring.

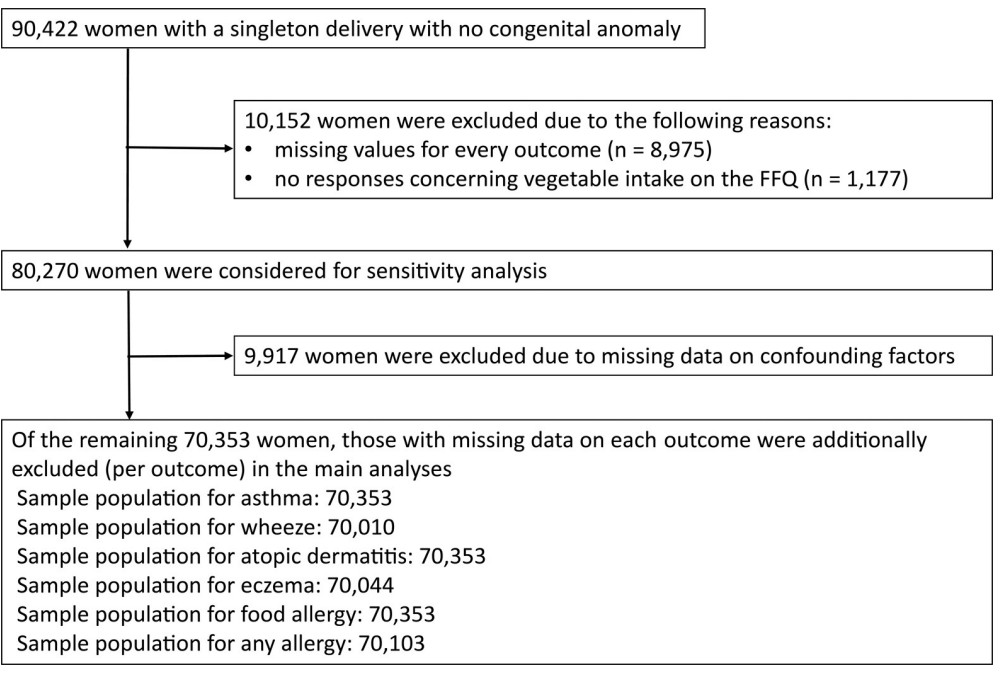

**Fig 1. Flow chart showing the study population selection.**

All statistical analyses were conducted using the statistical software package R version 3.5.2 for Windows.

## Results

The demographics of the mothers and their children are described in **Table 1**. Of the 80,270 participants, 2,027 (2.5%), 15,617 (19.6%), 3,477 (4.3%), 14,929 (18.7%), 13,801 (17.2%), and 25,028 (31.3%) children experienced asthma, wheeze, atopic dermatitis, eczema, food allergy, and any allergy, respectively. A small percentage of the participants had missing values.

The association between maternal dietary intake of vegetables and allergic outcomes is shown in **Table 2**. In terms of allergic outcomes in offspring at one year of age, all the aORs per quintile for each of the vegetables examined were close to 1.0, compared to the lowest quintile. For instance, women with the highest intake of total vegetables had a similar risk for asthma (aOR: 1.17, 95% CI: 0.88–1.20), wheeze (aOR: 1.02, 95% CI: 0.95–1.08), atopic dermatitis (aOR: 1.17. 95% CI: 1.04–1.31), eczema (aOR: 1.12, 95% CI: 1.05–1.19), and food allergy (aOR: 1.09, 95% CI: 1.02–1.16) as women with the lowest intake. Similar aORs were observed for other vegetable subclass categories, including folate vegetables, green and yellow vegetables, and cruciferous vegetables. Although some of those associations were statistically significant, the aORs ranged from 0.82 to 1.17, compared with the lowest quintile.

The association between maternal dietary intake of nutrients related to vegetables and allergic outcomes is shown in **Table 3**. As with vegetable intake, the aORs per quintile of maternal nutrient intake for all allergic outcomes were close to 1.0 (range: 0.87 to 1.13) compared with the lowest quintile. Crude and multivariate analyses showed a similar association in terms of both vegetables and related nutrients. Detailed data on the number of each quintile of vegetables and nutrients for the outcomes of interest are shown in **S1 Table**. The results of the crude analysis are shown in **S2 Table**.

**Table 1. Demographic background of the 80,270 participants.**

| Variables | | | Missing data (%) | Mean (SD) or N (%) |
|---|---|---|---|---|
| **Maternal demographics** | | | | |
| | Maternal age | | 11 (0.0) | 31.1 (5.0) |
| | Maternal height | | 10 (0.0) | 158.1 (5.3) |
| | Pre-pregnancy maternal weight | | 32 (0.0) | 53.0 (8.7) |
| | Maternal weight gain | | 613 (0.8) | 10.3 (4.0) |
| | Primaparae | | 344 (0.4) | 46,034 (57.3) |
| | ART pregnancy | | 308 (0.4) | 2,554 (3.2) |
| | Pre-existing hypertension | | 308 (0.4) | 342 (0.4) |
| | Pre-existing diabetes | | 308 (0.4) | 158 (0.2) |
| | Maternal allergic history | | 308 (0.4) | 40,834 (51.1) |
| | Maternal current smoking | | 846 (1.1) | 3,243 (4.1) |
| | Paternal current smoking | | 1,836 (2.3) | 35,916 (45.8) |
| | Cesarean section | | 138 (0.2) | 14,571 (18.2) |
| | Maternal education | | 281 (0.4) | |
| | | Junior High School | | 3,372 (4.2) |
| | | Senior High School | | 58,802 (73.3) |
| | | University | | 17,815 (22.2) |
| | Annual household income | | 5,243 (6.5) | |
| | | < 4 million-yen | | 29,630 (36.9) |
| | | ≥ 4 and < 6 million-yen | | 25,074 (31.2) |
| | | ≥ 6 million-yen | | 20,323 (25.3) |
| | Folic acid supplementation | | 517 (0.6) | 39650 (49.4) |
| | During pregnancy | | | |
| | Breastfeeding at one month after delivery | | 2203 (2.7) | |
| | | Breastfeeding only | | 42157 (52.5) |
| | | Mixed feeding | | 33039 (41.2) |
| | | Artificial milk feeding | | 2871 (3.6) |
| **Children's demographics** | | | | |
| | Allergic disorder at one year of age | | | |
| | | Asthma | 0 (0.0) | 2,027 (2.5) |
| | | Wheeze | 438 (0.5) | 15,617 (19.6) |
| | | Atopic dermatitis | 0 (0.0) | 3,477 (4.3) |
| | | Eczema | 387 (0.5) | 14,929 (18.7) |
| | | Food allergy | 0 (0.0) | 13,801 (17.2) |
| | | Any allergy | 311 (0.4) | 25,028 (31.3) |
| | Infant sex | | 16 (0.0) | |
| | | Male | | 41,003 (51.1) |
| | | Female | | 39,251 (48.9) |

The results of a sensitivity analysis using multiple imputations for the missing data are shown in **S3 Table**. The association observed in the sensitivity analysis was similar to that observed in the main analysis of cases with complete data. The results of the additional analysis of the association between maternal allergic history and vegetable intake are shown in **S4 Table**. There was no substantial difference in maternal allergic history among the categories when assessed by maternal vegetable intake.

**Table 2. Association between maternal vegetable intake and allergic disease in offspring.**

| Maternal intake[a] | Asthma | Wheeze | Atopic dermatitis | Eczema | Food allergy | Any allergy |
|---|---|---|---|---|---|---|
| | aOR[b] (95% CI) | aOR (95% CI) | aOR (95% CI) | aOR (95% CI) | aOR (95% CI) | aOR (95% CI) |
| **Total vegetables (g/day)** | | | | | | |
| Q1 (55.7±33.8) | reference | reference | reference | reference | reference | reference |
| Q2 (94.9±37.5) | 1.07 (0.92–1.24) | 1.00 (0.95–1.07) | 1.09 (0.97–1.22) | 1.02 (0.96–1.09) | 1.13 (1.06–1.20) | 1.06 (1.00–1.11) |
| Q3 (125.5±47.0) | 1.02 (0.87–1.19) | 0.99 (0.94–1.06) | 0.93 (0.82–1.05) | 1.00 (0.94–1.06) | 1.05 (0.98–1.12) | 1.01 (0.96–1.07) |
| Q4 (160.7±67.7) | 1.12 (0.87–1.19) | 1.01 (0.95–1.07) | 1.09 (0.97–1.23) | 1.08 (1.02–1.15) | 1.12 (1.05–1.19) | 1.09 (1.03–1.15) |
| Q5 (264.2±184.7) | 1.02 (0.88–1.20) | 1.02 (0.95–1.08) | 1.17 (1.04–1.31) | 1.12 (1.05–1.19) | 1.09 (1.02–1.16) | 1.08 (1.03–1.14) |
| **Folate vegetables (g/day)** | | | | | | |
| Q1 (9.0±6.4) | reference | reference | reference | reference | reference | reference |
| Q2 (24.1±16.1) | 0.93 (0.80–1.08) | 0.98 (0.93–1.05) | 1.01 (0.90–1.13) | 0.99 (0.93–1.05) | 1.02 (0.96–1.08) | 0.99 (0.94–1.05) |
| Q3 (27.5±10.5) | 0.89 (0.77–1.04) | 1.02 (0.96–1.09) | 0.95 (0.85–1.07) | 1.00 (0.94–1.06) | 1.00 (0.94–1.07) | 0.97 (0.92–1.02) |
| Q4 (38.4±14.9) | 0.98 (0.84–1.13) | 1.03 (0.97–1.09) | 1.05 (0.93–1.18) | 1.03 (0.97–1.10) | 1.00 (0.94–1.06) | 1.01 (0.96–1.06) |
| Q5 (69.0±54.1) | 0.87 (0.74–1.02) | 0.99 (0.93–1.06) | 1.11 (0.99–1.25) | 1.07 (1.01–1.14) | 1.02 (0.95–1.08) | 1.01 (0.96–1.06) |
| **Green and yellow vegetables (g/day)** | | | | | | |
| Q1 (15.5±10.0) | reference | reference | reference | reference | reference | reference |
| Q2 (30.5±12.4) | 0.99 (0.86–1.15) | 1.01 (0.95–1.07) | 1.02 (0.91–1.15) | 1.01 (0.95–1.08) | 1.06 (0.99–1.13) | 1.00 (0.95–1.06) |
| Q3 (43.0±16.5) | 0.98 (0.84–1.14) | 1.03 (0.97–1.10) | 1.03 (0.91–1.16) | 1.03 (0.97–1.10) | 1.08 (1.02–1.16) | 1.05 (1.00–1.11) |
| Q4 (57.9±24.1) | 0.96 (0.82–1.12) | 1.00 (0.94–1.07) | 1.07 (0.95–1.20) | 1.10 (1.04–1.17) | 1.12 (1.05–1.20) | 1.09 (1.03–1.15) |
| Q5 (102.1±73.4) | 0.95 (0.81–1.11) | 1.01 (0.95–1.08) | 1.11 (0.99–1.25) | 1.11 (1.05–1.19) | 1.11 (1.04–1.18) | 1.09 (1.03–1.15) |
| **Cruciferous vegetables (g/day)** | | | | | | |
| Q1 (10.3±6.6) | reference | reference | reference | reference | reference | reference |
| Q2 (21.1±8.0) | 0.82 (0.70–0.96) | 1.02 (0.96–1.08) | 0.93 (0.83–1.04) | 1.00 (0.94–1.06) | 1.01 (0.95–1.07) | 0.98 (0.93–1.04) |
| Q3 (30.4±11.6) | 0.95 (0.82–1.11) | 1.02 (0.96–1.08) | 1.00 (0.89–1.12) | 1.00 (0.94–1.07) | 1.00 (0.94–1.07) | 0.98 (0.93–1.03) |
| Q4 (42.4±15.0) | 0.93 (0.80–1.08) | 1.00 (0.94–1.06) | 1.01 (0.90–1.13) | 1.02 (0.96–1.08) | 0.98 (0.92–1.04) | 0.98 (0.93–1.03) |
| Q5 (75.2±56.4) | 0.83 (0.71–0.97) | 1.00 (0.94–1.06) | 1.01 (0.90–1.13) | 1.05 (0.98–1.11) | 0.96 (0.90–1.03) | 0.98 (0.93–1.03) |

aOR, adjusted odds ratio; CI, confidence interval

a: Q1 is the lowest quintile. Q5 is the highest quintile.

b: Adjusted for maternal age, place of recruitment, maternal height, pre-pregnant BMI, maternal weight gain during pregnancy, parity, conception method (assisted reproductive technology or not), pre-existing maternal hypertension, pre-existing maternal diabetes, parental allergic history, parental smoking, maternal education, maternal household income, infant gender, maternal folic acid supplementation during pregnancy, breastfeeding at one month after delivery, estimated maternal total energy intake during pregnancy, and delivery mode (vaginal or cesarean)

## Discussion

This study found no clear association between the maternal intake of vegetables and related nutrients with the development of asthma or wheezing among offspring at one year of age. Furthermore, our study demonstrated that neither factor contributed to any difference in the risk of atopic dermatitis, eczema, or food allergy.

The present study is the largest of its kind to examine the association between maternal vegetable intake during pregnancy and the development of allergic diseases in offspring at one year of old. While previous studies on the association of maternal vegetable intake with asthma or wheezing in offspring have yielded inconsistent results [9–14, 29, 30], our study, using a larger sample, was unable to find substantial differences in the risk of allergic disease development. One reason for this discrepancy might be the difference in the timeframes for assessing maternal dietary intake, as several studies have shown that maternal exposure during specific pregnancy periods is associated with allergic outcomes in offspring [9, 31–33]. Although a study on Japanese women showed that a significant, inverse association between maternal

**Table 3. Association between maternal nutritional intake and allergic diseases in offspring.**

| Maternal intake[a] | Asthma | Wheeze | Atopic dermatitis | Eczema | Food allergy | Any allergy |
|---|---|---|---|---|---|---|
| | aOR[b] | aOR | aOR | aOR | aOR | aOR |
| **Vitamin A (µg/day)** | | | | | | |
| Q1 (240.5±19.3) | reference | reference | reference | reference | reference | reference |
| Q2 (353.0±187.4) | 0.91 (0.78–1.05) | 0.97 (0.91–1.03) | 0.89 (0.79–1.00) | 1.00 (0.94–1.06) | 1.09 (1.02–1.16) | 1.04 (0.99–1.10) |
| Q3 (449.7±247.1) | 0.87 (0.76–1.03) | 0.98 (0.92–1.04) | 0.90 (0.80–1.01) | 1.04 (0.98–1.10) | 1.07 (1.01–1.14) | 1.06 (1.00–1.11) |
| Q4 (584.4±330.6) | 0.89 (0.76–1.03) | 0.99 (0.93–1.05) | 0.89 (0.80–1.00) | 1.03 (0.97–1.10) | 1.08 (1.01–1.15) | 1.05 (1.00–1.11) |
| Q5 (1053.6±1265.4) | 0.88 (0.76–1.02) | 1.02 (0.96–1.08) | 0.93 (0.83–1.05) | 1.03 (0.97–1.10) | 1.00 (0.94–1.07) | 1.01 (0.96–1.07) |
| **Alpha-carotene (µg/day)** | | | | | | |
| Q1 (102.8±81.4) | reference | reference | reference | reference | reference | reference |
| Q2 (255.7±106.7) | 1.01 (0.87–1.18) | 0.98 (0.92–1.04) | 0.99 (0.88–1.11) | 1.04 (0.98–1.11) | 1.07 (1.01–1.14) | 1.05 (1.00–1.11) |
| Q3 (377.4±190.7) | 1.00 (0.86–1.17) | 1.04 (0.98–1.11) | 0.95 (0.85–1.07) | 1.00 (0.94–1.06) | 1.10 (1.03–1.17) | 1.03 (0.98–1.09) |
| Q4 (581.8±215.6) | 1.09 (0.93–1.27) | 1.03 (0.97–1.10) | 1.00 (0.89–1.12) | 1.08 (1.02–1.15) | 1.08 (1.01–1.15) | 1.10 (1.04–1.16) |
| Q5 (1061.2±948.3) | 0.95 (0.81–1.11) | 0.99 (0.93–1.05) | 1.03 (0.92–1.16) | 1.10 (1.03–1.16) | 1.10 (1.04–1.18) | 1.09 (1.03–1.15) |
| **Beta-carotene (µg)/day** | | | | | | |
| Q1 (1127.6±771.4) | reference | reference | reference | reference | reference | reference |
| Q2 (1966.5±895.1) | 0.89 (0.77–1.04) | 1.01 (0.95–1.08) | 0.94 (0.83–1.06) | 1.01 (0.95–1.07) | 1.06 (1.00–1.13) | 1.01 (0.96–1.07) |
| Q3 (2678.3±1247.3) | 1.01 (0.87–1.17) | 1.05 (0.98–1.11) | 1.00 (0.89–1.12) | 1.01 (0.95–1.08) | 1.08 (1.01–1.15) | 1.05 (0.99–1.10) |
| Q4 (3508.8±1398.7) | 0.96 (0.83–1.12) | 1.02 (0.96–1.08) | 1.00 (0.89–1.13) | 1.08 (1.01–1.14) | 1.07 (1.01–1.15) | 1.07 (1.02–1.13) |
| Q5 (5753.5±4278.8) | 0.94 (0.81–1.10) | 1.02 (0.96–1.09) | 1.05 (0.94–1.18) | 1.08 (1.02–1.15) | 1.06 (0.99–1.13) | 1.06 (1.00–1.11) |
| **Vitamin C (mg/day)** | | | | | | |
| Q1 (39.0±25.2) | reference | reference | reference | reference | reference | reference |
| Q2 (61.5±26.8) | 1.00 (0.86–1.16) | 1.05 (0.99–1.11) | 0.97 (0.86–1.09) | 1.07 (1.01–1.14) | 1.07 (1.00–1.14) | 1.06 (1.01–1.12) |
| Q3 (81.5±41.3) | 1.00 (0.86–1.17) | 1.05 (0.99–1.12) | 1.03 (0.91–1.15) | 1.03 (0.97–1.10) | 1.07 (1.01–1.14) | 1.05 (1.00–1.11) |
| Q4 (102.9±47.2) | 1.07 (0.92–1.25) | 1.04 (0.98–1.11) | 1.08 (0.96–1.22) | 1.07 (1.00–1.13) | 1.05 (0.99–1.12) | 1.06 (1.00–1.11) |
| Q5 (157.1±98.3) | 1.02 (0.88–1.19) | 1.08 (1.01–1.14) | 1.10 (0.98–1.23) | 1.08 (1.01–1.15) | 1.00 (0.93–1.06) | 1.04 (0.98–1.09) |
| **Alpha-tocopherol (mg/day)** | | | | | | |
| Q1 (4.2±2.3) | reference | reference | reference | reference | reference | reference |
| Q2 5.4±2.3) | 1.01 (0.87–1.16) | 1.01 (0.95–1.07) | 1.03 (0.92–1.16) | 1.01 (0.95–1.07) | 1.02 (0.96–1.09) | 1.00 (0.95–1.05) |
| Q3 (6.2±2.5) | 1.03 (0.89–1.19) | 1.01 (0.95–1.07) | 1.07 (0.96–1.21) | 1.02 (0.96–1.08) | 1.04 (0.97–1.10) | 1.03 (0.98–1.08) |
| Q4 (7.1±3.2) | 0.88 (0.76–1.04) | 1.00 (0.94–1.07) | 1.03 (0.92–1.16) | 1.05 (0.99–1.12) | 1.07 (1.00–1.13) | 1.04 (0.99–1.10) |
| Q5 (9.3±5.5) | 0.89 (0.76–1.04) | 1.02 (0.96–1.08) | 1.08 (0.96–1.21) | 1.05 (0.99–1.12) | 1.05 (0.98–1.12) | 1.01 (0.96–1.07) |
| **Vitamin K (µg/day)** | | | | | | |
| Q1 (82.2±50.4) | reference | reference | reference | reference | reference | reference |
| Q2 (129.7±60.9) | 1.06 (0.91–1.23) | 1.04 (0.98–1.10) | 0.99 (0.89–1.12) | 1.03 (0.97–1.10) | 1.04 (0.98–1.11) | 1.03 (0.98–1.09) |
| Q3 (167.3±75.3) | 1.06 (0.91–1.24) | 1.01 (0.95–1.08) | 0.97 (0.86–1.09) | 1.01 (0.95–1.08) | 1.01 (0.95–1.08) | 1.01 (0.96–1.06) |
| Q4 (221.8±105.1) | 1.00 (0.85–1.16) | 1.03 (0.97–1.10) | 1.10 (0.98–1.24) | 1.03 (0.97–1.10) | 1.04 (0.98–1.11) | 1.02 (0.97–1.07) |
| Q5 (358.9±255.6) | 1.02 (0.87–1.19) | 1.00 (0.94–1.07) | 0.98 (0.87–1.10) | 1.02 (0.96–1.09) | 0.99 (0.92–1.05) | 0.98 (0.93–1.04) |
| **Folate (µg/day)** | | | | | | |
| Q1 (158.2±873.9) | reference | reference | reference | reference | reference | reference |
| Q2 (210.6±92.4) | 0.88 (0.76–1.03) | 1.02 (0.96–1.08) | 0.92 (0.82–1.04) | 1.06 (1.00–1.13) | 1.02 (0.96–1.08) | 1.03 (0.98–1.08) |
| Q3 (247.8±100.4) | 0.96 (0.83–1.12) | 1.02 (0.96–1.08) | 0.99 (0.88–1.12) | 1.08 (1.02–1.15) | 1.02 (0.96–1.09) | 1.03 (0.98–1.09) |
| Q4 (293.8±134.1) | 0.99 (0.85–1.15) | 1.03 (0.97–1.10) | 1.05 (0.93–1.17) | 1.06 (1.00–1.13) | 0.99 (0.93–1.05) | 1.02 (0.97–1.08) |
| Q5 (394.6±230.0) | 0.93 (0..80–1.09) | 1.07 (1.00–1.13) | 1.04 (0.93–1.17) | 1.08 (1.01–1.15) | 0.96 (0.90–1.03) | 1.00 (0.95–1.05) |
| **Soluble fiber (g/day)** | | | | | | |
| Q1 (1.5±0.7) | reference | reference | reference | reference | reference | reference |
| Q2 (2.0±0.7) | 0.98 (0.85–1.14) | 1.02 (0.96–1.08) | 0.98 (0.87–1.11) | 1.03 (0.97–1.10) | 1.04 (0.98–1.11) | 1.04 (0.98–1.09) |
| Q3 (2.4±0.9) | 0.96 (0.82–1.12) | 1.00 (0.94–1.06) | 1.09 (0.97–1.22) | 1.10 (1.04–1.17) | 0.94 (0.89–1.01) | 1.00 (0.95–1.06) |

*(Continued)*

**Table 3.** (Continued)

| Maternal intake[a] | Asthma | Wheeze | Atopic dermatitis | Eczema | Food allergy | Any allergy |
|---|---|---|---|---|---|---|
| | aOR[b] | aOR | aOR | aOR | aOR | aOR |
| Q4 (2.9±1.0) | 1.04 (0.90–1.21) | 1.04 (0.98–1.11) | 1.09 (0.97–1.23) | 1.09 (1.03–1.16) | 1.02 (0.96–1.09) | 1.05 (1.00–1.11) |
| Q5 (4.3±2.6) | 1.01 (0.86–1.18) | 1.01 (0.94–1.07) | 1.09 (0.97–1.22) | 1.14 (1.07–1.21) | 1.02 (0.95–1.09) | 1.07 (1.01–1.12) |
| **Insoluble fiber (g/day)** | | | | | | |
| Q1 (5.0±2.3) | reference | reference | reference | reference | reference | reference |
| Q2 (6.2±2.3) | 0.95 (0.82–1.11) | 0.97 (0.92–1.03) | 1.00 (0.89–1.13) | 1.07 (1.00–1.13) | 1.02 (0.96–1.08) | 1.04 (0.99–1.09) |
| Q3 (7.3±2.6) | 1.01 (0.87–1.18) | 1.00 (0.94–1.06) | 0.98 (0.87–1.11) | 1.07 (1.00–1.13) | 0.99 (0.93–1.05) | 1.03 (0.98–1.08) |
| Q4 (8.6±3.2) | 1.00 (0.86–1.16) | 1.00 (0.94–1.06) | 1.08 (0.96–1.21) | 1.11 (1.04–1.18) | 0.99 (0.93–1.06) | 1.05 (0.99–1.10) |
| Q5 (11.8±6.8) | 0.98 (0.83–1.14) | 1.01 (0.95–1.07) | 1.08 (0.96–1.21) | 1.12 (1.05–1.19) | 1.00 (0.93–1.06) | 1.04 (0.99–1.10) |
| **Total fiber (g/day)** | | | | | | |
| Q1 (6.8±3.2) | reference | reference | reference | reference | reference | reference |
| Q2 (8.7±3.3) | 1.01 (0.87–1.18) | 1.01 (0.96–1.08) | 1.03 (0.92–1.16) | 1.07 (1.01–1.14) | 1.04 (0.97–1.10) | 1.06 (1.01–1.12) |
| Q3 (10..3±3.9) | 0.97 (0.83–1.13) | 1.01 (0.95–1.08) | 1.01 (0.90–1.14) | 1.06 (1.00–1.13) | 0.96 (0.91–1.03) | 1.02 (0.97–1.07) |
| Q4 (12.0±4.5) | 1.05 (0.90–1.22) | 1.00 (0.94–1.06) | 1.09 (0.97–1.22) | 1.10 (1.03–1.17) | 1.02 (0.96–1.08) | 1.06 (1.00–1.11) |
| Q5 (16.4±9.6) | 0.97 (0.83–1.14) | 1.03 (0.96–1.09) | 1.08 (0.96–1.21) | 1.13 (1.06–1.20) | 1.01 (0.95–1.08) | 1.06 (1.01–1.12) |

a: Q1 is the lowest quintile. Q5 is the highest quintile.

b: Adjusted for maternal age, place of recruitment, maternal height, pre-pregnant BMI, maternal weight gain during pregnancy, parity, conception method (assisted reproductive technology or not), pre-existing maternal hypertension, pre-existing maternal diabetes, parental allergic history, parental smoking, maternal education, maternal household income, infant gender, maternal folic acid supplementation during pregnancy, breastfeeding at one month after delivery, estimated maternal total energy intake during pregnancy, and delivery mode (vaginal or cesarean)

vegetable intake and wheeze in offspring appeared only in early pregnancy [9], most studies did not focus on a specific timeframe for assessing exposure. However, a more likely explanation of these differences is the timing of the outcome assessments. Two studies on wheeze in offspring during the first year of life showed a non-significant association [12, 13], and some studies on older children showed a significant, inverse association [10, 29, 30], suggesting that maternal vegetable intake is not associated with asthma development during the first year of life. As infant wheezing, which is mainly caused by viral infections, and atopic asthma are uncommon in young infants, the present study's findings suggest that maternal vegetable intake may not be associated with the etiology of infant wheeze. However, wheeze and asthma at one year of age do not necessarily have the same etiology as when they appear later in life [34], leaving much yet to be clarified on the etiology of these conditions in children.

A few prospective studies that investigated the association between maternal vegetable intake during pregnancy and eczema in offspring produced inconsistent results [10, 11, 13]. Two studies attributed a significant association between them, while one study demonstrated that women with a higher vegetable intake during pregnancy were more likely to have offspring without atopic dermatitis at the age of 6.5 years [10]. Another study demonstrated a significant inverse association between maternal green and yellow vegetable intake during pregnancy and the development of eczema in offspring aged 16–24 months [11]. Two studies, one examining eczema development at one year of age [13] and the other examining eczema development at two years of age [14], showed a non-significant association between maternal vegetable intake and eczema development. Our results indicate the absence of a substantial risk for eczema development during the first year of life in offspring regardless of the category of maternal vegetable intake during pregnancy, suggesting that the association between maternal vegetable intake and eczema development depends on the age of the offspring, as seen in wheeze and asthma development in offspring at one year of age.

The present study also suggests that women with a higher intake of total vegetables, folate-rich vegetables, green and yellow vegetables, and certain nutrients have a slightly higher risk of their offspring developing eczema and food allergy. Although the reasons for this slightly elevated risk are difficult to ascertain; environmental factors, such as the presence of pesticide residues on produce grown in Japan, might provide an explanation. While no study has investigated the association between early-life exposure to pesticides and atopic dermatitis development in childhood, some studies have demonstrated a significant, positive association between pesticide exposure and the development of childhood asthma [35–37]. As childhood asthma, food allergy, and atopic dermatitis are part of the allergy march [38], atopic changes might be followed by food allergy and asthma as a consequence of prenatal exposure to unidentified environmental factors. Our results found no positive association between vegetable intake during pregnancy and asthma/wheeze or eczema in offspring; this may be due to the difficulty of assessing asthma outcomes at one year of age, as discussed above. Thus, future research on these topics is required.

Most previous studies on vitamin A, C, and E reported a non-significant association with allergic outcomes [4], while one study reported a significant inverse association between maternal vitamin E intake and childhood wheeze [39]. Studies of maternal folate intake have shown mixed results on its association with wheeze and asthma [40, 41], and none have demonstrated a significant association with eczema or atopic dermatitis [42, 43]. Our study demonstrated that maternal vitamin and folate intake did not have a substantial association with allergic outcomes, including asthma, wheeze, eczema, and food allergy, in offspring. The relatively young age of the offspring examined in this study may be another reason for our findings. On the other hand, no studies have yet investigated the possibility of an association between maternal dietary fiber intake and allergic disease in offspring, despite a recent study suggesting that the metabolism of dietary fiber may influence the development of allergic diseases of the airway [44]. At any rate, the current study found no substantial association, in line with studies of other nutrients.

The strength of our study includes the large sample size, which enabled the accurate assessment of the association between dietary intake and allergic outcomes as well as sufficient adjustment for potential confounders. We also confirmed the validity of the FFQ used in the current study because the mean EI/BMR in our samples was 1.52 (data not shown) [45]. Furthermore, although our sample included cases with missing information, a sensitivity analysis with an imputed dataset was used to confirm the accuracy of the data. We believe that this study presents a reliable result on the association between maternal vegetable intake during pregnancy and allergic outcomes in offspring at one year of age. Nonetheless, this study has several limitations. First, allergies appearing at one year of age do not always continue into later life [34, 46] despite allergic disorders at this age being a risk factor for subsequent allergic disease development. Further longitudinal studies investigating the association between maternal vegetable intake during pregnancy and the development of allergic disease in offspring using the same birth cohort over a longer follow-up period are warranted. Second, the FFQ used in the current study was validated for use with the general population but not with pregnant women [21, 22]. Furthermore, misclassification of dietary intake, which could lead to underestimation, may have occurred as the FFQ was self-reported. Third, although previous studies have used asthma, atopic dermatitis, and food allergy based on the doctor's diagnosis as outcome variables [47, 48], these have not been validated. However, we complementally used those variables in addition to wheeze and eczema using a questionnaire because the assessment of allergic outcomes in offspring at one year of age is challenging. There may also have been some misclassification risk using two tools for the outcome assessment (self-reported questionnaire and doctor's diagnosis).

## Conclusions

The present study demonstrated non-substantial differences in the risk of asthma, wheeze, atopic dermatitis, eczema, and food allergy development in offspring at one year of age due to maternal intake of vegetables and related nutrients.

## Supporting information

**S1 Table. Number of each quintile of vegetables and nutrients for the outcomes of interest.** (DOCX)

**S2 Table. Crude odds ratio of the allergic outcomes between quintiles for each exposure category.** (DOCX)

**S3 Table. Association between maternal nutritional intake and allergic disease in offspring based on the imputed data set.** (DOCX)

**S4 Table. Relationship between maternal vegetable intake and maternal allergic history.** (DOCX)

## Acknowledgments

We are grateful to all the participants and individuals involved in the data collection. We would also like to thank the following members of the JECS as of 2019: Michihiro Kamijima (principal investigator, Nagoya City University, Nagoya, Japan), Shin Yamazaki (National Institute for Environmental Studies, Tsukuba, Japan), Yukihiro Ohya (National Center for Child Health and Development, Tokyo, Japan), Reiko Kishi (Hokkaido University, Sapporo, Japan), Nobuo Yaegashi (Tohoku University, Sendai, Japan), Koichi Hashimoto (Fukushima Medical University, Fukushima, Japan), Chisato Mori (Chiba University, Chiba, Japan), Shui-chi Ito (Yokohama City University, Yokohama, Japan), Zentaro Yamagata (University of Yamanashi, Chuo, Japan), Hidekuni Inadera (University of Toyama, Toyama, Japan), Takeo Nakayama (Kyoto University, Kyoto, Japan), Hiroyasu Iso (Osaka University, Suita, Japan), Masayuki Shima (Hyogo College of Medicine, Nishinomiya, Japan), Youichi Kurozawa (Tot-tori University, Yonago, Japan), Narufumi Suganuma (Kochi University, Nankoku, Japan), Koichi Kusuhara (University of Occupational and Environmental Health, Kitakyushu, Japan), and Takahiko Katoh (Kumamoto University, Kumamoto, Japan). We would also like to thank James R. Valera of the Department of Education for Clinical Research of the National Center for Child Health and Development for his assistance in editing this manuscript.

## Author Contributions

**Conceptualization:** Kohei Ogawa, Kyongsun Pak.

**Data curation:** Kohei Ogawa, Kyongsun Pak.

**Formal analysis:** Kohei Ogawa.

**Methodology:** Kohei Ogawa, Kyongsun Pak.

**Supervision:** Kohei Ogawa, Kiwako Yamamoto-Hanada, Kazue Ishitsuka, Hatoko Sasaki, Hidetoshi Mezawa, Mayako Saito-Abe, Miori Sato, Limin Yang, Minaho Nishizato, Mizuho Konishi, Haruhiko Sago, Yukihiro Ohya.

**Validation:** Kohei Ogawa, Kazue Ishitsuka, Hatoko Sasaki, Hidetoshi Mezawa, Mayako Saito-Abe, Miori Sato, Limin Yang, Minaho Nishizato, Mizuho Konishi, Haruhiko Sago, Yukihiro Ohya.

**Writing – original draft:** Kohei Ogawa.

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
