## [Decision Letter · Decision Letter 0]

7 Oct 2020

PONE-D-20-22340

Association between maternal vegetable intake during pregnancy and allergy in offspring: Japan Environment and Children's Study (JECS).

PLOS ONE

Dear Dr. kohei,

Thank you for submitting your manuscript to PLOS ONE. After careful consideration, we feel that it has merit but does not fully meet PLOS ONE’s publication criteria as it currently stands. Therefore, we invite you to submit a revised version of the manuscript that addresses the points raised during the review process.

We look forward to receiving your revised manuscript.

Kind regards,

Calistus Wilunda, DrPH

Academic Editor

PLOS ONE

Journal Requirements:

2.We note that you have indicated that data from this study are available upon request. PLOS only allows data to be available upon request if there are legal or ethical restrictions on sharing data publicly. For more information on unacceptable data access restrictions, please see http://journals.plos.org/plosone/s/data-availability#loc-unacceptable-data-access-restrictions.

3. One of the noted authors is a group or consortium [Japan Environment and Children’s Study (JECS) Group^]. In addition to naming the author group, please list the individual authors and affiliations within this group in the acknowledgments section of your manuscript. Please also indicate clearly a lead author for this group along with a contact email address.

Reviewers' comments:

Reviewer's Responses to Questions

**Comments to the Author**

1. Is the manuscript technically sound, and do the data support the conclusions?

Reviewer #1: Yes

Reviewer #2: Yes

Reviewer #3: Yes

2. Has the statistical analysis been performed appropriately and rigorously? 

Reviewer #1: Yes

Reviewer #2: Yes

Reviewer #3: Yes

3. Have the authors made all data underlying the findings in their manuscript fully available?

Reviewer #1: No

Reviewer #2: Yes

Reviewer #3: Yes

4. Is the manuscript presented in an intelligible fashion and written in standard English?

Reviewer #1: Yes

Reviewer #2: Yes

Reviewer #3: Yes

5. Review Comments to the Author

Reviewer #1: This novel paper investigates the potential association between maternal intake of vegetables during pregnancy and the risk of a range of childhood allergic diseases. The paper seek to add new knowledge to inconsistent results on this specific topic being conducting analyses in a large well conducted and well described young cohort. The authors apply appropriate statistical models and present a range of results, which are presented in a simple concise way. However, the authors could consider an alternative presentation of results in table 3. I have only a range of minor comments and some considerations, which could be added to the discussion.

Minor comments

Abstract:

Please add in relation to the aOR, 95% confidence intervals (95%CI) in the methods, and the results. And add the 95%CI to the presentation of results.

Introduction:

Who has stated that “maternal vegetable intake during pregnancy is one of the most interesting exposures? Please, reconsider the use of “most interesting”

It would be relevant to add a few words on proposed mechanisms of the anti-oxidative and anti-inflammatory properties in relation to development of immune responses potentially working during fetal life – which later maybe related to development of allergic diseases.

Please, add to the description of previous studies, the type of studies and range of n in the “relatively small sample sizes”.

Did any of the studies showing a positive association between vegetables and allergic diseases use register data or self-reported questionnaire data? Please, add a bit more information on these studies which can be used for discussion later.

Materials and methods:

Line 89, In the method section, please add information in which gestational weeks the FFQ was completed.

Line 95-97: it is unclear whether women who were missing information in one variable only was excluded or not. Is it correct those women who were missing values on every outcome, were excluded – not if they were missing values in one or two, then they were kept in the analyses? Nevertheless, in line 97 the analyses used a complete dataset – meaning no missing values at all? Maybe change wording to ‘missing values on one or more outcomes’ if that were the case.

Line 98: please add number of observations in the multiple imputation data set.

Line 102: is it possible to add a reference to the “other nationally valid regulations”?

Line 106: please add mean gestational weeks of when the FFQ was completed during pregnancy.

Line 107: what does the authors mean by ‘other profiles’? Please, add sufficient information in the text.

Regarding vegetables groups: Please, explain the rationale for having spinach in two groups. What are the pros and cons? Please, if any cons could influence the observed associations or interpretations of results, this should be included in the discussion.

Line 125: Please, add one sentence on the validity of the ISSAC tool

Line 130: has the doctor’s diagnosis been validated at some point? How well is coverage of cases using the doctor diagnosis?

Is there a risk of misclassification using these two tools for outcome assessments?

Line 142: Please, add a sentence that the covariates were selected based on previous literature, which seems to be the case.

Line 146+148: there are some uncertainity about the numbers. The remaining 80,270 do have missing data, not on all outcomes but maybe some? Please, add n for the dataset used for the amin analysis with ‘women with no missing data’ and those ‘in the multiple imputation sensitivity analyses’

The wording “ a quintile base on the energy-adjusted estimate of the intake for total vegetables” does not seem correct and is difficult to understand. Please reconsider this sentence and title for Figure 1.

Table 1. In some lines, n(%) has been added to the test e.g. “Junior high school” but not by others e.g. “maternal allergic history” etc. Please, consider to be more consistent.

Maybe say “Allergic disorder at age 1” instead of “at endpoint”

Line 184: “for other kinds of vegetables” are to unspecific. Which vegetables are the authors referring to?

Table 2 and 3. Please, try to make each column fit the text in a way that the results aOR (95%CI) can fit in one line. Makes it easier to read.

In the foot note, has ‘ART’ been written fully. I think, the authors should also write fully in a foot note.

Please, add that the unit is per day? Vitamin A ug/day. Also please, add n.

Would it be possible to makes figures or the like of (some) results in table 3? This tables is rather long across several pages.

The discussion:

The authors, briefly discuss the timing of the FFQ in relation to observed findings. However, is the timing relevant in relation to the development of auto immune responses in fetal life? And is the timing irrelevant since ‘most other studies did not focus on the time frame’?

The authors could discuss in more depth the ‘difficulties in assessing the asthma outcomes at age 1. Please see my questions above on the validity of outcome assessment, proc&cons

Did any of the studies showing a positive association between vegetables and allergic diseases use register data or self-reported questionnaire data?

How did the present study obtain the goal of being able to add significant knowledge to the existing knowledge base with this larger cohort study?

Line 240: what is the prevalence of wheeze and asthma in older children?

Line 277: does the authors mean? “using similar birth cohorts” or “using the same birth cohort with longer follow-up”?

How would misclassification of dietary intake by the FFQ affect the observed associations? Please, add a sentence to the discussion.

Reviewer #2: Dear authors,

This is a very interesting paper investigating the associations between maternal diet during pregnancy and allergic diseases in the offspring at age 1.

My main concern is the fact that the analysis were performed without adjusting on maternal breastfeeding, which can be a vety important factor in allergic diseases.

I would also suggest to adjust on maternal supplementation during pregnancy.

My minor concern is : page 7, line 88 : please delete "the" before maternal.

Reviewer #3: This paper aim at assessing the association between maternal intake of vegetables and related nutrients during pregnancy with allergic diseases in offspring at age 1 year. In order to reach this aim the authors use the information provided by a cohort study on a large sample of pregnant women. The information on dietary exposures was collected by means of a Food Frequency Questionnaire, whereas health outcomes in the offspring at age 1 year through the International Study of Asthma and Allergies in Childhood questionnaire.

The authors state that maternal intake of vegetables and other related nutrients during pregnancy had little or no association with the considered health outcomes in offspring at age 1 year.

The paper is well written and the authors have in general used proper methods to analyze their data. However, I would like to seek clarification on some points.

- Did the author perform some form of quality check on the FFQ questionnaire (for example: did they drop subjects with implausible values of estimated energy? Did they evaluate the ratio of energy intake to basal metabolic rate?)?

- Due to the poor quality of the image, I could not properly understand the utility of figure 1. What does figure 1 add to the text to explain how the authors created energy-adjusted quintiles of maternal vegetable intakes/related nutrients? Maybe some important details of the statistical model could be enlightened by this figure.

- The authors consider several covariates in the multivariate models. Among these covariates total energy intake is not present. Why? Even when the residual method is used, it is generally recommended to include total energy intake as a covariate in the model (see 1. Willet WC. Nutritional epidemiology 2nd ed. New York: Oxford Univercity Press; 1998; pag.275; 2. Willett W, Stampfer MJ. Total energy intake, implications for epidemiologic analyses. Am J Epidemiol. 1986;124:17‐27.)

- The approach of the authors in considering the results is cautious. In fact, even if they found that some of the adjusted measures of association between quintiles of the considered dietary exposures and health outcomes were significantly different from 1.0 when compared with the lowest quintile, they chose not to consider this evidence as a straightforward clue of true association. This choice is related to the fact that the estimated adjusted ORs for the association between the dietary exposures and the considered health outcomes are close to 1. This interpretation is quite reasonable. Nevertheless, in the discussion, the authors discuss some of their results, i.e. the slightly higher risk of eczema development in the offspring for women with a higher intake of vegetables and of certain nutrients. Why the author focused their attention only on eczema? Actually, the present study suggests also that women with a higher intake of total vegetables, folate rich vegetables, green and yellow vegetables, and certain nutrients have a slightly higher risk of food allergy and of other allergies. If they think it appropriate to discuss the results concerning eczema, they should discuss the results concerning food allergy and other allergies as well.

6. PLOS authors have the option to publish the peer review history of their article (what does this mean?). If published, this will include your full peer review and any attached files.

Reviewer #1: No

Reviewer #2: No

Reviewer #3: No

---

## [Author Response · Author response to Decision Letter 0]

10 Dec 2020

Reviewer #1

 This novel paper investigates the potential association between maternal intake of vegetables during pregnancy and the risk of a range of childhood allergic diseases. The paper seek to add new knowledge to inconsistent results on this specific topic being conducting analyses in a large well conducted and well described young cohort. The authors apply appropriate statistical models and present a range of results, which are presented in a simple concise way. However, the authors could consider an alternative presentation of results in table 3. I have only a range of minor comments and some considerations, which could be added to the discussion.

Minor comments

Abstract:

Please add in relation to the aOR, 95% confidence intervals (95%CI) in the methods, and the results. And add the 95%CI to the presentation of results.

Thank you for pointing this out. We have added 95% confidence intervals in the methods and results sections as detailed below.

Lines 36-39

Using the categorizations as exposure variables, the adjusted odds ratios (aOR) with 95% confidence intervals (CI) were determined for the allergic outcomes, including asthma, wheeze, atopic dermatitis, eczema, and food allergy, in the offspring per quintile at one year of age.

Lines 43-46

The lowest aOR was found in the association of maternal cruciferous vegetable intake with asthma (aOR: 0.82, 95% CI: 0.70–0.96), and the highest was shown in the association of maternal total vegetable intake with atopic dermatitis (aOR: 1.17, 95% CI: 1.04–1.31).

Introduction:

Who has stated that “maternal vegetable intake during pregnancy is one of the most interesting exposures? Please, reconsider the use of “most interesting”

It would be relevant to add a few words on proposed mechanisms of the anti-oxidative and anti-inflammatory properties in relation to development of immune responses potentially working during fetal life – which later maybe related to development of allergic diseases.

Thank you very much for this suggestion. We have removed "most interesting" and added a few words on the proposed mechanisms as detailed below.

Lines 60-62

Maternal vegetable intake during pregnancy is an interesting exposure category due to the anti-oxidative and anti-inflammatory properties of vegetables, which may be related to the development of allergic diseases.

Please, add to the description of previous studies, the type of studies and range of n in the “relatively small sample sizes”.

Thank you for pointing this out. We have added the type of studies and the range of the sample as below.

Lines 68-70

One possible explanation for this inconsistency is the relatively small sample sizes (ranging from 310 to 3086) used in these observational studies.

Did any of the studies showing a positive association between vegetables and allergic diseases use register data or self-reported questionnaire data? Please, add a bit more information on these studies which can be used for discussion later.

Thank you very much for this comment. As all of the studies with a significant association were based on self-reported data, we have included this information in our revised manuscript, as detailed below.

Lines 62-64

Although several studies of maternal vegetable intake during pregnancy have investigated its association with the development of allergic disease in offspring using a self-reported database, they have yielded inconsistent results. 

Materials and methods:

Line 89, In the method section, please add information in which gestational weeks the FFQ was completed.

Thank you for pointing this out. The FFQ was completed by the participant at any time during their pregnancy. We have added the gestational age when the FFQ was completed as follows.

Lines 86-88

A self-administered food frequency questionnaire (FFQ), which was completed during pregnancy (as a general rule, from 22+0 to 27+6), was used to assess the maternal diet during pregnancy.

Line 95-97: it is unclear whether women who were missing information in one variable only was excluded or not. Is it correct those women who were missing values on every outcome, were excluded – not if they were missing values in one or two, then they were kept in the analyses? Nevertheless, in line 97 the analyses used a complete dataset – meaning no missing values at all? Maybe change wording to ‘missing values on one or more outcomes’ if that were the case.

We apologize for the confusion. Our sample population included women with information about their vegetable intake on the FFQ and with at least one information on allergic outcomes. The missing number for each outcome is described in Table 1. We have revised this part as follows: 

Lines 92-97

Women with a singleton delivery without congenital malformations were included. Women who were missing values on every outcome variable or failed to supply information about their vegetable intake on the FFQ were excluded (those missing one or two values were kept in our study population). Of those, our primary analysis was conducted based on the dataset with complete data on each variable, as our dataset contained a substantial sample.

Line 98: please add number of observations in the multiple imputation data set.

We apologize for this confusion. We have described our sample population in the statistical section as below. 

Lines 148-156

Of the 90,422 women with a singleton delivery without a congenital malformation, 8,975 and 1,177 were excluded due to missing values for every outcome and no responses concerning vegetable intake on the FFQ, respectively. Thus, the remaining 80,270 women were included in our study population. Our main analysis was conducted after excluding those with missing data on confounding factors (n = 9,917) and each outcome (n for wheeze: 438, n for eczema: 387, n for any allergy: 311 [some of those overlapped with missing confounding factors]). Thus, the main analyses for asthma, wheeze, atopic dermatitis, eczema, food allergy, and any allergy were conducted for 70,353, 70,010, 70,353, 70,044, 70,353, and 70,103 people, respectively. A sensitivity analysis on 80,270 women using multiple imputations was conducted for the entire cohort (Fig 1).

Line 102: is it possible to add a reference to the “other nationally valid regulations”?

Thank you for this suggestion. We have added a reference in this sentence.

Lines 100-101

The JECS was conducted in accordance with the guidelines of the Declaration of Helsinki and other nationally valid regulations [19].

Line 106: please add mean gestational weeks of when the FFQ was completed during pregnancy.

Thank you very much for this important suggestion, and we agree with the importance of including the mean gestational weeks when the FFQ was completed. However, the gestational week dataset contains some missing or non-reliable data, and we were not able to accurately calculate the mean gestational age when the FFQ was completed. Thus, we can only state that the FFQ was completed by participants at some point during pregnancy, without the mean gestational age.

Lines 86-88

A self-administered food frequency questionnaire (FFQ), which was completed during pregnancy (as a general rule, from 22+0 to 27+6), was used to assess the maternal diet during pregnancy.

Line 107: what does the authors mean by ‘other profiles’? Please, add sufficient information in the text.

We apologize for this inaccurate sentence. We have revised as below.

Lines 106-107

The FFQ was completed during mid- to late-pregnancy, and other profiles of the FFQ have been described elsewhere in detail.

Regarding vegetables groups: Please, explain the rationale for having spinach in two groups. What are the pros and cons? Please, if any cons could influence the observed associations or interpretations of results, this should be included in the discussion.

Thank you for this comment. We agree that many nutritional studies have categorized food groups and nutritional groups by mutual exclusion. However, we aimed to determine how various kinds of vegetables are associated with allergic outcomes in offspring (based on a hypothesis that those categories may have different effects). The same categorization used in the current study has also been applied previously: PMID: 29434319). Folate-rich vegetables, green and yellow vegetables, and cruciferous vegetables are easy to understand categories, and both involve spinach. Considering our concept of this study, we believe that the current categorization is adequate. We have stated that our categorization was based on a previous study, as follows.

Lines 114-115

Vegetables were categorized into subclasses according to the following definitions, according to a previous study [9].

Line 125: Please, add one sentence on the validity of the ISSAC tool

Thank you very much for pointing this out. We have added a sentence on the validation as follows:

Lines 125-127

Wheezing and eczema symptoms in the offspring at this age were assessed using a questionnaire, which was modified from the International Study of Asthma and Allergies in Childhood (ISSAC) as a validated questionnaire.

Line 130: has the doctor’s diagnosis been validated at some point? How well is coverage of cases using the doctor diagnosis? Is there a risk of misclassification using these two tools for outcome assessments?

Thank you for pointing this out. This tool was not validated, and thus there could be a risk of misclassification. The coverage of cases was unknown. We have stated these as limitations of the current study, as below.

Lines 298-303

Third, although previous studies have used asthma, atopic dermatitis, and food allergy based on the doctor’s diagnosis as outcome variables [47, 48], these have not been validated. However, we complementally used those variables in addition to wheeze and eczema using a questionnaire because the assessment of allergic outcomes in offspring at one year of age is challenging. There may also have been some misclassification risk using two tools for the outcome assessment (self-reported questionnaire and doctor’s diagnosis).

Line 142: Please, add a sentence that the covariates were selected based on previous literature, which seems to be the case.

Thank you for this advice. We have revised as below.

Lines 144-145

The covariates were treated as confounding factors according to previous studies.

Line 146+148: there are some uncertainity about the numbers. The remaining 80,270 do have missing data, not on all outcomes but maybe some? Please, add n for the dataset used for the amin analysis with ‘women with no missing data’ and those ‘in the multiple imputation sensitivity analyses’

We apologize for this inaccurate description. The main analysis was based on women without missing confounding factors (n = 9,917) and each outcome. We have revised this part as below and added a Figure (flow chart).

Lines 148-157

Of the 90,422 women with a singleton delivery without a congenital malformation, 8,975 and 1,177 were excluded due to missing values for every outcome and no responses concerning vegetable intake on the FFQ, respectively. Thus, the remaining 80,270 women were included in our study population. Our main analysis was conducted after excluding those with missing data on confounding factors (n = 9,917) and each outcome (n for wheeze: 438, n for eczema: 387, n for any allergy: 311 [some of those overlapped with missing confounding factors]). Thus, the main analyses for asthma, wheeze, atopic dermatitis, eczema, food allergy, and any allergy were based on 70,353, 70,010, 70,353, 70,044, 70,353, and 70,103 people, respectively. A sensitivity analysis based on 80,270 women using multiple imputations was conducted for the entire cohort (Fig 1).

Fig 1: Flow chart showing the study population selection 

The wording “ a quintile base on the energy-adjusted estimate of the intake for total vegetables” does not seem correct and is difficult to understand. Please reconsider this sentence and title for Figure 1.

We apologize for this incorrect sentence. However, as we noticed that this figure contained small information, thus we have decided to remove this figure and the relevant sentence.

An example of a quintile based on the energy-adjusted estimate of the total vegetable intake is shown in Fig 1.

Fig1: An example of a quintile based on the energy-adjusted intake for total vegetables

Table 1. In some lines, n(%) has been added to the test e.g. “Junior high school” but not by others e.g. “maternal allergic history” etc. Please, consider to be more consistent.

Maybe say “Allergic disorder at age 1” instead of “at endpoint”

Thank you for pointing this out. We have revised Table 1 accordingly.

Line 184: “for other kinds of vegetables” are to unspecific. Which vegetables are the authors referring to?

Thank you very much for pointing this out. We have revised this sentence to be more specific as below.

Lines 197-198

Similar aORs were observed for other vegetable subclass categories, including folate vegetables, green and yellow vegetables, and cruciferous vegetables.

Table 2 and 3. Please, try to make each column fit the text in a way that the results aOR (95%CI) can fit in one line. Makes it easier to read. 

In the foot note, has ‘ART’ been written fully. I think, the authors should also write fully in a foot note.

Please, add that the unit is per day? Vitamin A ug/day. Also please, add n.

Thank you for this suggestion. We have revised Tables 2 and 3 to make them easier to read. We have changed ART to assisted reproductive technology in a foot note.

We have added that the unit is per day. 

As Tables 2 and 3 are busy in their current form, we have created a separate supplemental table showing the number (S1 Table).

Line 207-208

Detailed data on the number of each quintile of vegetables and nutrients for the outcomes of interest are shown in S1 Table.

Would it be possible to makes figures or the like of (some) results in table 3? This tables is rather long across several pages.

Thank you for this suggestion. We have carefully considered how we could make them smaller (e.g., figures). However, we did not find a good solution. If we create a figure instead of Table 3, this may be inconsistent with Table 2. Thus, we propose that the current presentation in Table 3 may be better. We have tried to make it easier to read by making each column fit the text.

 Even after taking these changes into consideration, Table 3 might be better presented as a supplemental table if the reviewer recommends this. However, as we believe the results shown in Table 3 are important, we hope to leave Table 3 in the main manuscript if possible.

The discussion:

The authors, briefly discuss the timing of the FFQ in relation to observed findings. However, is the timing relevant in relation to the development of auto immune responses in fetal life? And is the timing irrelevant since ‘most other studies did not focus on the time frame’?

Thank you very much for this comment. While the timing of the development of auto immune responses in fetal life is still unknown, several studies have shown that maternal exposure during specific pregnancy periods is associated with allergic outcomes in offspring. Thus, we have added a sentence explaining why we considered timing as a potential reason for these discrepancies.

Lines 227-230

One reason for this discrepancy might be the difference in the timeframes for assessing maternal dietary intake, as several studies have shown that maternal exposure during specific pregnancy periods is associated with allergic outcomes in offspring [9, 31-33].

The authors could discuss in more depth the ‘difficulties in assessing the asthma outcomes at age 1. Please see my questions above on the validity of outcome assessment, proc&cons

Thank you very much for this advice. We have discussed this in the limitation section, as follows.

Lines 298-303

Third, although previous studies have used asthma, atopic dermatitis, and food allergy based on the doctor’s diagnosis as outcome variables [47, 48], these have not been validated. However, we complementally used those variables in addition to wheeze and eczema using a questionnaire because the assessment of allergic outcomes in offspring at one-year-old is challenging. There may also have been some misclassification risk using two tools for the outcome assessment (self-reported questionnaire and doctor’s diagnosis).

Did any of the studies showing a positive association between vegetables and allergic diseases use register data or self-reported questionnaire data?

Thank you very much for this comment. As all of the studies with a significant association were based on self-reported data, we have added this information as below.

Lines 62-64

Although several studies of maternal vegetable intake during pregnancy have investigated its association with the development of allergic disease in offspring using a self-reported database, they have yielded inconsistent results. 

How did the present study obtain the goal of being able to add significant knowledge to the existing knowledge base with this larger cohort study?

Thank you very much for this comment. We have added further detail on this as below.

Lines 284-290

The strength of our study includes the large sample size, which enabled the accurate assessment of the association between dietary intake and allergic outcomes as well as sufficient adjustment for potential confounders. We also confirmed the validity of the FFQ used in the current study because the mean EI/BMR in our samples was 1.52 (data not shown) [45]. Furthermore, although our sample included cases with missing information, a sensitivity analysis with an imputed dataset was used to confirm the accuracy of the data. We believe that the current study presents a reliable result on the association between maternal vegetable intake during pregnancy and allergic outcomes in offspring at one year of age. 

Line 240: what is the prevalence of wheeze and asthma in older children?

We apologize for this confusion. Our intention with this sentence (as seen in wheeze and asthma development) is to highlight the non-significant association between maternal vegetable intake during pregnancy with wheeze and asthma in offspring at one year of age. Thus, we have revised this part as below.

Lines 251-255

Our results indicate the absence of a substantial risk for eczema development during the first year of life in offspring regardless of the category of maternal vegetable intake during pregnancy, suggesting that the association between maternal vegetable intake and eczema development depends on the age of the offspring, as seen in wheeze and asthma development in offspring at one year of age.

Line 277: does the authors mean? “using similar birth cohorts” or “using the same birth cohort with longer follow-up”?

Thank you very much for pointing this out. We have revised this as follows.

Lines 293-295

Further longitudinal studies investigating the association between maternal vegetable intake during pregnancy and the development of allergic disease in offspring using the same birth cohort over a longer follow-up period are warranted. 

How would misclassification of dietary intake by the FFQ affect the observed associations? Please, add a sentence to the discussion.

Thank you very much for pointing this out. We have clarified the possible effect of this misclassification on the observed associations.

Lines 297-298

Furthermore, misclassification of dietary intake, which could lead to underestimation, may have occurred as the FFQ was self-reported.  

Reviewer #2

-

Dear authors,

This is a very interesting paper investigating the associations between maternal diet during pregnancy and allergic diseases in the offspring at age 1.

My main concern is the fact that the analysis were performed without adjusting on maternal breastfeeding, which can be a vety important factor in allergic diseases.

I would also suggest to adjust on maternal supplementation during pregnancy.

Thank you for this important suggestion. We added maternal breastfeeding and maternal folic acid supplementation during pregnancy (other supplementation is not popular for pregnant women in Japan) as confounders. Furthermore, we have added total energy intake as a confounder, according to other reviewers’ comments.

We have revised the method section as below.

Lines 140-144

Other variables, such as the place of recruitment, parental smoking status, maternal SES, including maternal education (junior high school, senior high school, university), household income (<4 million-yen, ≥4 and <6 million-yen, ≥6 million-yen), maternal folic acid supplementation during pregnancy (yes or no), and breastfeeding at one month after delivery (breastfeeding only, mixed feeding, artificial mild feeding), were obtained via a questionnaire at baseline.

Lines 165-171

For multivariate analysis, maternal age, place of recruitment, maternal height, pre-pregnancy body mass index (BMI), maternal weight gain during pregnancy, parity, conception method (assisted reproductive technology or not), pre-existing maternal hypertension, pre-existing maternal diabetes, parental allergic history, parental smoking, maternal education, maternal household income, infant gender, maternal folic acid supplementation during pregnancy, breastfeeding at one month after delivery, estimated maternal total energy intake during pregnancy, and delivery mode were adjusted.

My minor concern is : page 7, line 88 : please delete "the" before maternal.

Thank you for pointing this out. We deleted "the" as the reviewer commented.

 

Reviewer #3:

This paper aim at assessing the association between maternal intake of vegetables and related nutrients during pregnancy with allergic diseases in offspring at age 1 year. In order to reach this aim the authors use the information provided by a cohort study on a large sample of pregnant women. The information on dietary exposures was collected by means of a Food Frequency Questionnaire, whereas health outcomes in the offspring at age 1 year through the International Study of Asthma and Allergies in Childhood questionnaire.

The authors state that maternal intake of vegetables and other related nutrients during pregnancy had little or no association with the considered health outcomes in offspring at age 1 year.

The paper is well written and the authors have in general used proper methods to analyze their data. However, I would like to seek clarification on some points.

- Did the author perform some form of quality check on the FFQ questionnaire (for example: did they drop subjects with implausible values of estimated energy? Did they evaluate the ratio of energy intake to basal metabolic rate??

Thank you for this suggestion. We calculated the ratio of energy intake to the basal metabolic rate (EI/BMR) to evaluate the quality of the FFQ. As the mean EI/BMR in our samples was 1.52, we believe that the FFQ used in this study was adequate. We have stated this in the strength section as below.

Line 286-287

We also confirmed the validity of the FFQ used in the current study because the mean EI/BMR in our samples was 1.52 (data not shown) [45].

- Due to the poor quality of the image, I could not properly understand the utility of figure 1. What does figure 1 add to the text to explain how the authors created energy-adjusted quintiles of maternal vegetable intakes/related nutrients? Maybe some important details of the statistical model could be enlightened by this figure.

Thank you for this comment. As we noticed that this figure contained small information, we decided to remove this figure and the corresponding sentence.

An example of a quintile based on the energy-adjusted estimate of the total vegetable intake is shown in Fig 1.

Fig1: An example of a quintile based on the energy-adjusted intake for total vegetables

- The authors consider several covariates in the multivariate models. Among these covariates total energy intake is not present. Why? Even when the residual method is used, it is generally recommended to include total energy intake as a covariate in the model (see 1. Willet WC. Nutritional epidemiology 2nd ed. New York: Oxford Univercity Press; 1998; pag.275; 2. Willett W, Stampfer MJ. Total energy intake, implications for epidemiologic analyses. Am J Epidemiol. 1986;124:17‐27.)

Thank you for this important suggestion. We have added the total energy intake as the reviewer pointed out. Furthermore, we have added maternal breastfeeding and maternal folic acid supplementation during pregnancy as confounders, according to the other reviewers' comments.

We have revised the method section as below.

Lines 165-171

For multivariate analysis, maternal age, place of recruitment, maternal height, pre-pregnancy body mass index (BMI), maternal weight gain during pregnancy, parity, conception method (assisted reproductive technology or not), pre-existing maternal hypertension, pre-existing maternal diabetes, parental allergic history, parental smoking, maternal education, maternal household income, infant gender, maternal folic acid supplementation during pregnancy, breastfeeding at one month after delivery, estimated maternal total energy intake during pregnancy, and delivery mode were adjusted. 

- The approach of the authors in considering the results is cautious. In fact, even if they found that some of the adjusted measures of association between quintiles of the considered dietary exposures and health outcomes were significantly different from 1.0 when compared with the lowest quintile, they chose not to consider this evidence as a straightforward clue of true association. This choice is related to the fact that the estimated adjusted ORs for the association between the dietary exposures and the considered health outcomes are close to 1. This interpretation is quite reasonable. Nevertheless, in the discussion, the authors discuss some of their results, i.e. the slightly higher risk of eczema development in the offspring for women with a higher intake of vegetables and of certain nutrients. Why the author focused their attention only on eczema? Actually, the present study suggests also that women with a higher intake of total vegetables, folate rich vegetables, green and yellow vegetables, and certain nutrients have a slightly higher risk of food allergy and of other allergies. If they think it appropriate to discuss the results concerning eczema, they should discuss the results concerning food allergy and other allergies as well.

Thank you for pointing this out. We agree that we should discuss the results concerning food allergies. Because food allergy is a part of the allergy march, the hypothesis of this association is the same as eczema. Thus, we did not add another explanation for food allergy but revised it from the previous manuscript. Although women with a higher intake of vegetables also had a slightly higher risk of “any allergy,” this association may be due to eczema and food allergy. Thus, we did not discuss “any allergy” in this paragraph.

Lines 257-269

The present study also suggests that women with a higher intake of total vegetables, folate-rich vegetables, green and yellow vegetables, and certain nutrients have a slightly higher risk of their offspring developing eczema and food allergy. Although the reasons for this slightly elevated risk are difficult to ascertain, environmental factors, such as the presence of pesticide residues on produce grown in Japan, might provide an explanation. While no study has investigated the association between early-life exposure to pesticides and atopic dermatitis development in childhood, some studies have demonstrated a significant, positive association between pesticide exposure and the development of childhood asthma [35-37]. As childhood asthma, food allergy, and atopic dermatitis are part of the allergy march [38], atopic changes might be followed by food allergy and asthma as a consequence of prenatal exposure to unidentified environmental factors. Our results found no positive association between vegetable intake during pregnancy and asthma/wheeze or eczema in offspring; this may be due to the difficulty of assessing asthma outcomes at one year of age, as discussed above. Thus, future research on these topics is required.

---

## [Decision Letter · Decision Letter 1]

8 Jan 2021

Association between maternal vegetable intake during pregnancy and allergy in offspring: Japan Environment and Children’s Study

PONE-D-20-22340R1

Dear Dr. kohei,

We’re pleased to inform you that your manuscript has been judged scientifically suitable for publication and will be formally accepted for publication once it meets all outstanding technical requirements.

Kind regards,

Calistus Wilunda, DrPH

Academic Editor

PLOS ONE

Additional Editor Comments (optional):

Reviewers' comments:

Reviewer's Responses to Questions

**Comments to the Author**

1. If the authors have adequately addressed your comments raised in a previous round of review and you feel that this manuscript is now acceptable for publication, you may indicate that here to bypass the “Comments to the Author” section, enter your conflict of interest statement in the “Confidential to Editor” section, and submit your "Accept" recommendation.

Reviewer #1: All comments have been addressed

Reviewer #2: All comments have been addressed

Reviewer #3: All comments have been addressed

2. Is the manuscript technically sound, and do the data support the conclusions?

Reviewer #1: Yes

Reviewer #2: Yes

Reviewer #3: Yes

3. Has the statistical analysis been performed appropriately and rigorously? 

Reviewer #1: Yes

Reviewer #2: Yes

Reviewer #3: Yes

4. Have the authors made all data underlying the findings in their manuscript fully available?

Reviewer #1: No

Reviewer #2: Yes

Reviewer #3: Yes

5. Is the manuscript presented in an intelligible fashion and written in standard English?

Reviewer #1: Yes

Reviewer #2: Yes

Reviewer #3: Yes

6. Review Comments to the Author

Reviewer #1: The authors have sufficiently replied to my previous questions and i believe the paper is ready for publication. The supplementary table 1 works fine, I think. I have only a few minor comments for the authors to consider.

Minors

Lines 306-307: “ Furthermore, misclassification of dietary intake, which could lead to underestimation, may have occurred as the FFQ was self-reported.”

Does the authors mean “underestimation of the observed associations”? please, make the sentence a bit more clear/specific on what would be the consequence of misclassification of diet intake on the observed associations.

Line 236: One reason for this discrepancy might be the difference in the timeframes for assessing maternal dietary intake, as several studies have shown that maternal exposure during specific pregnancy periods is associated with allergic outcomes in offspring [9, 31-33].

The authors added an important consideration, however, it would give even more information if “the specific weeks of gestation” could be mentioned:

“One reason for this discrepancy might be the difference in the timeframes for assessing maternal dietary intake, as several studies have shown that maternal exposure during specific pregnancy periods (e.i. GW xx-xx) is associated with allergic outcomes in offspring [9, 31-33].”

The wording seems a bit incorrect: “S1 Table. Number of each quintile of vegetables and nutrients for the outcomes of interest” Should this be changed to: “S1 Table. Number of cases per quintile of vegetables and nutrients for each outcomes of interest” ? Should be revised in the manuscript accordingly.

This title of S3 should be revised to S4: “S4 Table. Relation between maternal vegetable intake and maternal allergic history”

In addition, does the table show relation or prevalence?

Reviewer #2: (No Response)

Reviewer #3: I think that the authors have adequately addressed my previous comments, in general. The changes made in response to the reviewers improved the manuscript. As a side comment, I would point out that when I asked some additional information on the evaluation of the quality of food frequency data, I expected the authors to make changes to the analysis dataset, for example by eliminating subjects with extreme values of EI, and/or of EI/BMR. In the modified version of the paper, the authors state that the average value of EI/BMR is 1.55, this does not exclude the existence of subjects with extreme and implausible values of that ratio. Even if I am not completely satisfied with this specific answer, nonetheless I believe that the methods used in the paper are in general valid and appropriate.

7. PLOS authors have the option to publish the peer review history of their article (what does this mean?). If published, this will include your full peer review and any attached files.

Reviewer #1: No

Reviewer #2: No

Reviewer #3: No

---

## [Editor Report · Acceptance letter]

14 Jan 2021

PONE-D-20-22340R1 

Association between maternal vegetable intake during pregnancy and allergy in offspring: Japan Environment and Children’s Study 

Dear Dr. Ogawa:

I'm pleased to inform you that your manuscript has been deemed suitable for publication in PLOS ONE. Congratulations! Your manuscript is now with our production department. 

Kind regards, 

on behalf of

Dr. Calistus Wilunda 

Academic Editor

PLOS ONE